# Oxidative Stress and Preeclampsia-Associated Prothrombotic State

**DOI:** 10.3390/antiox9111139

**Published:** 2020-11-17

**Authors:** Cha Han, Pengzhu Huang, Meilu Lyu, Jingfei Dong

**Affiliations:** 1Department of Obstetrics and Gynecology, Tianjin Medical University General Hospital, Tianjin 300052, China; tjhancha@tmu.edu.cn (C.H.); huangpz2019@163.com (P.H.); lyumeilu@tmu.edu.cn (M.L.); 2Tianjin Key Laboratory of Female Reproductive Health and Eugenic, Tianjin 300052, China; 3Bloodworks Research Institute, Seattle, WA 98102, USA; 4Division of Hematology, Department of Medicine, School of Medicine, University of Washington, Seattle, WA 98102, USA

**Keywords:** preeclampsia, hemostasis, platelets, coagulation, oxidative stress

## Abstract

Preeclampsia (PE) is a common obstetric disease characterized by hypertension, proteinuria, and multi-system dysfunction. It endangers both maternal and fetal health. Although hemostasis is critical for preventing bleeding complications during pregnancy, delivery, and post-partum, PE patients often develop a severe prothrombotic state, potentially resulting in life-threatening thrombosis and thromboembolism. The cause of this thrombotic complication is multi-factorial, involving endothelial cells, platelets, adhesive ligands, coagulation, and fibrinolysis. Increasing evidence has shown that hemostatic cells and factors undergo oxidative modifications during the systemic inflammation found in PE patients. However, it is largely unknown how these oxidative modifications of hemostasis contribute to development of the PE-associated prothrombotic state. This knowledge gap has significantly hindered the development of predictive markers, preventive measures, and therapeutic agents to protect women during pregnancy. Here we summarize reports in the literature regarding the effects of oxidative stress and antioxidants on systemic hemostasis, with emphasis on the condition of PE.

## 1. Introduction

Preeclampsia (PE) is a gestational disease that severely endangers maternal and fetal health and can develop into more severe complications (e.g., eclampsia) with long-term consequences. PE is defined as newly onset hypertension or proteinuria after 20 weeks of gestation without a prior history of hypertension, or newly onset hypertension with any of the following systemic manifestations: thrombocytopenia, renal insufficiency, impaired liver function, pulmonary edema, and severe headache without alternative diagnoses or visual impairments [1]. As the leading cause of maternal and perinatal morbidity and mortality, PE affects approximately 2% to 8% of pregnancies worldwide [2]. In an age-adjusted cohort study, the rate of PE in the US increased from 3.4% in 1980 to 3.8% in 2010 but the rate of severe PE increased by 322% during the same period [3]. This drastic increase was believed to be caused primarily by the age-cohort effect but it also highlights the need for improving pregnancy care. The current treatment of PE in clinical practice is delivery in time but antihypertension and spasmolysis may be used to control the progression of PE, prevent severe complications like eclampsia, and prolong the gestational week to improve maternal and fetal survival.

The maternal complications of PE include cerebrovascular diseases, acute renal failure, and subcapsular hematoma of the liver; the adverse perinatal outcomes include preterm delivery, fetal growth restriction (FGR), and fetal death [4]. PE patients also have a higher risk for long-term complications such as hypertension, ischemic heart disease, stroke, and venous thromboembolism [5]. Women with normal pregnancies often develop a systemic hypercoagulable state that progresses into a prothrombotic state in PE patients. In this review, we summarize the current knowledge of the role of oxidative stress in the development of this PE-associated prothrombotic state. This review focuses on linking oxidative stress to the prothrombotic state associated with PE. While both conditions develop frequently in patients with PE, they have been studied individually, without taking their causal relationship into consideration. Studying both conditions together will allow us to define this causal relationship, thus developing more accurate predictive markers and new targeted therapeutics for the patients.

The literature search was conducted in PubMed of the National Library of Medicine using the following keywords, either alone or in different combinations: placenta development, preeclampsia, HELLP syndrome, platelets, endothelial cells, coagulation factors, fibrinolysis, oxidative stress, and antioxidants. These keywords identified 1091 articles, of which 169 were considered relevant. The findings from these relevant reports were reviewed and discussed in the manuscript.

## 2. Placental Development

The placenta serves as a transient organ supplying nutrients and oxygen to and removing wastes from the fetus through the umbilical cord. It consists of the basal decidua, chorion frondosum, and amnion. Before trophoblast invasion, the distal tips of the spiral arteries are obstructed by the endovascular trophoblast plugs, and the maternal arterial connection with the intervillous space is restricted to the network of tortuous intercellular space. As a result, blood in the intervillous space has low oxygen tension, which is necessary to protect embryos from free radical-induced damage [6,7,8]. After the spiral artery plugs are gradually loosened and removed, maternal arterial blood flows into the intervillous space, increasing the oxygen tension from <20 mmHg at 8 weeks of gestation to >50 mmHg at 12 weeks. This transition results in a burst of oxidative stress in the placenta [9].

During early placental development, cytotrophoblast cells either fuse into syncytiotrophoblasts, which are exposed to maternal blood for maternal–fetal exchange or invade the decidua and myometrium [10]. The balance between trophoblast proliferation and invasion is regulated by oxygen. Cytotrophoblast cells proliferate in 2% oxygen, which mimics the uterine environment before 10 weeks of gestation, but begin to differentiate at 20% oxygen [11]. Extravillous trophoblast cells have a higher invasive activity at 17% oxygen than at 3% oxygen [12]. If these punctual and stage-specific rates of oxygenation become dysregulated during early pregnancy, placental development can be disrupted or impaired, resulting in various pathologies including PE [13].

## 3. Pathogenesis of PE

Placental villous lesions are found in 45.2% of PE patients compared with 14.6% of women with normal pregnancies, with vascular lesions being most common [14]. However, maternal conditions such as vascular disease, obesity, and autoimmune disease also contribute substantially to poor placentation and remodeling of the spiral arteries, leading to placental tissue ischemia and oxidative stress [15]. Placental tissue hypoxia is a hallmark event of PE, characterized by the overexpression of hypoxia-inducible transcription factor (HIF) [16] and poor angiogenesis in the placenta. It is associated with hypertension, proteinuria, and FGR [17].

The poor placental angiogenesis can result from dysregulation of growth factors such as vascular endothelial cell growth factor (VEGF) and the associated intracellular signaling pathways. For example, VEGF binds the fms-like tyrosine kinase receptor (Flt) to trigger proangiogenic signals. However, PE patients have significantly enhanced expression of soluble Flt 1 (sFlt-1) [18], an alternatively spliced variant of Flt that lacks the transmembrane and intracellular domains [19]. sFlt-1 binds VEGF and placental growth factor (PLGF) with a high affinity to competitively block VEGF binding to membrane-bound Flt and the resulting angiogenesis [19]. Placental biopsies have indeed shown that placentas from the majority of PE patients have poor trophoblastic invasion and vasculopathy of the spiral arteries that includes fibrin deposition, acute atherosis, and thrombosis [20]. More importantly, these local placental lesions can disseminate systemically by releasing soluble factors and extracellular vesicles into the maternal circulation [21,22], resulting in systemic endothelial injury and a prothrombotic state, as seen in PE patients.

There are two forms of PE: early-onset PE develops prior to 34 weeks of gestation and late-onset PE occurs after 34 gestational weeks. While both forms of PE are caused by syncytiotrophoblast stress, early-onset PE is closely associated with insufficient remodeling of spiral arteries and the resulting poor placentation, whereas late-onset PE often results from the restriction of placenta growth [15]. Early-onset PE is associated with more fetal death (adjusted odds ratios (OR) 5.8; 95% CI 4.0–8.3 vs. adjusted OR 1.3; 95% CI 0.8–2.0) and perinatal death/morbidity (adjusted OR 16.4; 95% CI 14.5–18.6 vs. adjusted OR 2.0; 95% CI 1.8–2.3) than late-onset PE [23].

## 4. The PE-Associated Hypercoagulable State

Pregnant women, especially during their late pregnancies, develop a systemic hypercoagulable state that is physiologically important for preventing excessive bleeding during delivery and postpartum. The cause of this hypercoagulable state is multi-factorial. For example, the syncytiotrophoblast, which is directly exposed to maternal circulation, lacks proliferative capacity and is maintained by fusion of the underlying cytotrophoblast cells, with the assistance of syncytin. Syncytiotrophoblastic cells therefore undergo constant apoptosis [24,25,26]. These apoptotic cells express anionic phospholipids on their surface. Anionic phospholipids such as phosphatidylserine (PS) and phosphatidylethanolamine are normally present on the inner leaflet, whereas neutral phospholipids such as phosphatidylcholine (PC) and sphingomyelin are on the external leaflet of a cell membrane bilayer. This asymmetrical distribution of membrane phospholipids is maintained by enzymatic transporters [27,28] but is lost during apoptosis [29], leading to the exposure of PS on the outer membrane. The anionic phospholipids are highly procoagulant because they promote and accelerate the formation of tenase to generate thrombin [30]. Apoptotic syncytiotrophoblast cells can also shed from the placenta into the circulation to form large multinucleated syncytial knots, which not only initiate coagulation but also invoke the immune-mediated inflammatory response. These syncytial knots can be removed by macrophages [31]. In addition, the apoptotic syncytiotrophoblastic cells also release PS-exposed extracellular vesicles to the maternal circulation [21,22] to induce hypercoagulation and inflammation, as syncytial knots do.

This protective hypercoagulable state is significantly exaggerated in pregnant women with placenta-mediated complications, such as PE, becoming a pathological prothrombotic state or thrombosis [32]. Feldstein et al. [33] studied 348 PE patients and reported that 16.1% had abnormal coagulation profiles defined by changes in prothrombin time (PT), international normalization ratio (INR), activated partial thromboplastin time (aPTT), and/or plasma fibrinogen. However, the actual rate of coagulation dysregulation is likely to be significantly higher because these tests (1) are calibrated to detect bleeding diathesis and are insensitive for detecting prothrombotic states, and (2) could not accurately measure the procoagulant activity of activated platelets and endothelial cells. Consistent with this finding, Moldenhauer et al. [34] reported that PE patients have significantly higher rates of intervillous thrombi (OR 1.95; 95% CI 1.0–3.7), central tissue infarction (OR 5.9; 95% CI 3.1–11.1), and thrombi in the fetal circulation (OR 2.8; 95% CI 1.2–6.6) than women with normotensive pregnancies. In addition to PE, dysregulated hemostasis is also found in patients with hemolysis, elevated liver enzymes, and low platelet count (HELLP syndrome), which is considered to be a severe complication of PE, but its pathogenesis is not identical to that of PE [35]. Women with early-onset HELLP syndrome have a lower prevalence of proteinuria, hypertriglyceridemia, hyperglycemia, and thrombophilia but higher levels of the inflammatory mediator homocysteine than those with early-onset PE [36]. Patients with HELLP syndrome are therefore more likely to develop severe inflammation [37] and thus are likely to suffer more severe oxidative stress. Patients with PE and those with HELLP syndrome are known to be in a hypercoagulable state [38] but whether the hypercoagulable state differs in level and nature between the two conditions remains to be studied. There is no published report on whether oxidative stress differentially regulates the hypercoagulable state associated with PE and that with HELLP syndrome.

Multiple factors contribute to the transition from the hypercoagulable state found in most normal pregnancies to the prothrombotic state developed primarily in PE. For example, both tissue factor, which initiates the extrinsic coagulation following vascular injury, and plasminogen activator inhibitor (PAI)-1, which blocks fibrinolysis, are elevated in the blood and placental tissues of PE patients, likely due to injuries to trophoblast cells undergoing repeated cycles of hypoxia–reoxygenation [39]. PE patients also have an enhanced rate of thrombin generation [40], which proteolytically promotes fibrin formation and activates platelets and endothelial cells to express procoagulant activity [41,42,43]. Consistent with these observations, more activated platelets are detected in PE patients [44,45], which increases platelet–leukocyte aggregation in circulation, a common marker for the prothrombotic state in PE [46]. These platelet–leukocyte aggregates upregulate the expression of sFlt-1, propagating endothelial dysfunction and inflammation in PE [47]. We have recently shown that placenta-derived extracellular vesicles (pcEVs) are detectable in the peripheral blood of pregnant women but the levels are significantly higher in PE patients [22]. These pcEVs are highly procoagulant because they express anionic phospholipids such phosphatidylserine (PS) on their surfaces and are associated with the prothrombotic state of PE patients [22]. In mouse models, we further showed that these pcEVs induce hypertension and proteinuria not only in pregnant mice but also in non-pregnant mice [21], demonstrating the importance of pcEVs in the pathogenesis of PE. These pcEVs can cause PE through multiple pathways [48] by (1) promoting coagulation through PS exposed on their surfaces [21,22,49], (2) expressing tissue factor to trigger extrinsic coagulation [49], and (3) activating platelets and endothelial cells to release procoagulant extracellular vesicles [21]. Together, these findings demonstrate that PE is initiated by placental factors and propagated by maternal factors in the systemic proinflammatory and hypercoagulable states, which are closely associated with oxidative stress.

## 5. PE and Oxidative Stress

The placental hypoperfusion caused by poor transformation of the spiral arteries and the resulting ischemia and reperfusion damage can cause the severe oxidative stress found in placentas from PE patients. This oxidative stress develops because of an imbalance between the production and clearance of oxidants. Nicotinamide adenine dinucleotide phosphate oxidase (NADPH oxidase), which catalyzes the production of superoxide free radicals, is found to be hyperactive in placentas from PE patients, especially on the surface of the syncytiotrophoblast microvilli [50,51], suggesting that PE placentas produce more superoxide. In contrast, the NADPH oxidase isoform NOX4, which protects vascular function [52], is downregulated in the placental villous tissue of PE patients [53]. The activity of xanthine oxidase (XO), which uses O_2_ as the electron acceptor to generate reactive oxygen species (ROS) [54], is also higher in the plasma and cytotrophoblast cells of PE patients than women with normal pregnancies [55,56].

Mitochondria, which provide 90% of the cellular metabolic energy through aerobic energy production [57], can become a major source of intracellular and extracellular oxidative stress when they become dysfunctional, as found in the placentas of PE patients [58]. Placental ischemia reduces mitochondrial respiration, increasing ROS production during PE [59,60,61] and leading to mitochondrial oxidative stress in placental cells [6,60]. However, the mitochondrial oxidative stress is not limited to placental cells; it can also occur in maternal organs such as the kidneys [60]. In addition to intracellular mitochondrial dysfunction, morphologically intact extracellular mitochondria (exMTs) have been increasingly detected in diseased states [62,63,64,65,66]. We have recently shown that exMTs released during acute phase reactions remain metabolically competent and capable of generating ATP and ROS [67]. However, the evidence directly linking oxidative stress to the PE-induced prothrombotic state remains circumstantial, as we discuss in the following sections.

## 6. Oxidative Stress in the PE-Induced Prothrombotic State

Human hemostasis consists of four distinct but closely related components: endothelial cells, platelets and adhesive ligands, coagulation, and fibrinolysis (Figure 1). Their dysfunction, either alone or in different combinations, can result in bleeding diathesis or thrombosis in the veins and arteries. The PE-associated prothrombotic state involves all four components, but how oxidative stress modifies these hemostatic components remains poorly understood.

### 6.1. Oxidative Stress on the Endothelium

The intact endothelium lining the vascular lumen is an anticoagulant and antithrombotic surface because it is covered by a protective layer of proteoglycans called the glycocalyx [68]. The endothelium exposed to oxidative stress and inflammatory stimuli loses the glycocalyx and becomes highly procoagulant by increasing its permeability, reducing the bioavailability of the vascular relaxing factor nitro-oxide (NO) and the antiplatelet factor prostaglandin [69], tethering platelets and leukocytes to its surface [70], and promoting the deposition of fibrin generated through coagulation. Exogenous superoxide activates the endothelial cells to release proinflammatory mediators (e.g., interleukin (IL)-8, tumor necrosis factor (TNF)-α) and express adhesion molecules (e.g., Intercellular Adhesion Molecule 1 and P-selectin) and tissue factor [71,72].

### 6.2. Oxidative Stress on Platelets

Platelets adhere to and aggregate on the subendothelial matrix to form a hemostatic plug at the site of vascular injury. Their reactivity to common agonists is regulated by endogenous and exogenous redox signals [73]. Platelets produce and release ROS upon activation by common agonists [74] and can also be activated [75] or sensitized by ROS for activation by other platelet stimuli [76]. For example, the subendothelial matrix protein collagen stimulates platelets to produce ROS by binding to the glycoprotein (GP) VI and subsequent signaling [77,78]. Platelets produce ROS primarily through NAD(P)H-oxidase (NOX) after exposure to agonists; the ROS thus produced regulate the activation of αIIbβ3 integrin but have a minimal impact on the secretion from α- and dense-granules and the shape changes of platelets [74]. Human platelets express both NOX1 and NOX2 [79], which differentially regulate platelet reactivity [80,81]. Platelets produce O_2_^−^ when they are exposed to anoxia–reoxygenation conditions and aggregate [82]. Superoxide can be converted to hydrogen peroxide (H_2_O_2_), which also induces platelet aggregation [76] and enhances the aggregation induced by collagen and arachidonic acid [78,83]. H_2_O_2_ sensitizes platelets for collagen-induced activation by oxidizing thiols in the SH2 domain-containing protein tyrosine phosphatase-2 in the GP VI-mediated signaling pathway [78,84,85]. Platelet reactivity is attenuated by the redox factor NO [86] and inhibited by superoxide dismutase (SOD), catalase, and the hydroxyl radical scavengers mannitol and deoxyribose. The redox states of thiols in key platelet receptors also regulate the platelets’ reactivity to agonists [87]. For example, the platelet-derived protein disulfide isomerase (PDI), which is detected on the surface of resting platelets [88], regulates the redox states of active thiols on αIIbβ3 integrin for binding fibrinogen [89,90]. PDI also mediates the disulfide exchange required for collagen binding to its platelet receptor α2β1 [91]. It is therefore not surprising that platelets deficient in PDI and its homologs activate and aggregate poorly in response to agonists [92].

In addition to soluble oxidants, metabolically active exMTs released from injured cells can also become a significant and persistent source of oxidative stress [67] because blood is only mildly reducing compared with the highly reducing environment of the cytoplasm [93]. These exMTs bind platelets through the lipid receptor CD36 to activate them in an ROS-dependent manner [67]. By binding to target cells, ROS produced by exMTs can be localized, concentrated, and potentially shielded from the antioxidant activity of the blood.

Oxidative stress has been well established as a key pathology of PE but evidence of how oxidative stress regulates platelets in PE patients remains circumstantial. Although oxidized and free cysteine and homocysteine are significantly increased, the ratio of free to oxidized cysteine is lower in the blood of PE patients than in that of women with normotensive pregnancies [94], suggesting a systemic oxidative stress that could affect the redox state of platelet surface thiols [93]. Platelets from PE patients also have reduced L-arginine transport, increased protein carbonyl content, and reduced catalase activity, compared with women with normotensive pregnancies [95]. Consistent with this notion, we detected significantly more platelet-derived extracellular vesicles in PE patients than in women with normotensive pregnancies [22]. These EVs can be released from platelets exposed to oxidative stress [67,96,97]. This finding is reproducible in mouse models of PE [21]. Platelets activated by oxidative stress can be rapidly removed from circulation, resulting in thrombocytopenia, as reported for mice deficient in endothelial nitric oxide synthase (eNOS) in the presence of excess sFlt-1 [98]. Furthermore, patients with HELLP syndrome, a severe complication of PE [35], develop thrombocytopenia primarily due to thrombotic microangiopathy [99,100], which is similar to that found in patients with thrombotic thrombocytopenic purpura. This finding not only implicates platelets in the pathogenesis of PE and its complications but also suggests that von Willebrand factor (VWF), a key adhesive ligand that mediates platelet hemostasis [101], is dysfunctional in patients with PE and in those with HELLP syndrome.

### 6.3. Oxidative Stress on Platelet Adhesive Ligands

Oxidative stress regulates not only platelet reactivity but also the adhesive ligands that bind and activate platelets. Among these adhesive ligands, VWF has been studied most extensively for redox regulations. VWF is synthesized in megakaryocytes and endothelial cells as large multimers, which can interact simultaneously with multiple receptors on the same cell (receptor crosslinking) or different cells (cell coupling). Once synthesized, VWF multimers are either constitutively released into the circulation or stored in the Weibel–Palade bodies of endothelial cells and the α-granules of megakaryocytes/platelets, where multimerization continues to generate ultra-large (UL) VWF multimers [101]. These ULVWF multimers are highly prothrombotic because they are intrinsically hyperadhesive and spontaneously bind circulating platelets, endothelial cells, and other cells [102,103,104,105]. They are released when endothelial cells and platelets are activated by inflammatory and oxidative stimuli and are anchored to endothelial cells, where they are rapidly cleaved at the Y1605–M1606 peptide bond in the A2 domain of VWF by the metalloprotease ADAMTS-13 (a disintegrin and metalloprotease with a thrombospondin Type 1 motif, Member 13) into smaller multimers, which are then released into circulation [106,107,108]. These smaller circulating VWF multimers are significantly less adhesive (inert), binding platelets poorly unless they are immobilized on subendothelial collagen at the site of vascular injury or activated by high fluid shear stress [109]. However, oxidative stress can activate these inert VWF multimers or enhance their adhesive activity through several means. First, it oxidizes the methionine residues of VWF, especially at the cleavage site (M1606) to make it resistant to cleavage by ADAMTS-13 [110]. VWF oxidized by peroxynitrite is also resistant to ADAMTS-13 cleavage [111]. Second, multiple cysteine thiols on VWF can be oxidized to form laterally associated hyperadhesive fibrils [112,113]. Third, the oxidation of two vicinal cysteines in the A2 domain induces a conformational change to expose the platelet-binding A1 domain and activate VWF [114]. Fourth, methionine oxidation also makes ADAMTS-13 less active in cleaving VWF [115]. Finally, thiols at the *C*-terminus of ADAMTS-13 can be oxidized to abolish their disulfide bond-reducing activity [116].

While this has not been specifically reported, several lines of circumstantial evidence suggest that the activities of both VWF and ADAMTS-13 are changed in PE patients. First, systemic and placental oxidative stress has been demonstrated extensively in the condition of PE. Second, plasma VWF antigens and their adhesive activities are both significantly elevated in patients with PE and those with HELLP syndrome [22,117,118,119]. Third, plasma ADAMTS-13 antigens and activity are reduced in patients with PE or HELLP syndrome [117,120], resulting in a kinetic ADAMTS-13 deficiency and hyperadhesive VWF.

### 6.4. Oxidative Stress on Coagulation and Fibrinolysis

At the site of vascular injury, platelets adhere and aggregate on the subendothelial matrix to form a plug that seals the wound (Figure 1). This platelet plug is stabilized by crosslinked fibrin, which is generated from fibrinogen cleaved by thrombin. When hemostasis is achieved, the fibrin polymers are lysed by plasmin to reestablish circulation occluded by the hemostatic plug. Both coagulation and fibrinolysis are achieved through serial enzymatic reactions involving multiple coagulation and fibrinolytic factors.

Thrombin was reported to be inactivated irreversibly by increasing levels of oxygen in 1949 [121], providing the first evidence of oxidative regulation of coagulation. Both the intrinsic and extrinsic pathways are susceptible to shifts in the redox balance, with the intrinsic pathway being more sensitive to oxidation, whereas the extrinsic pathway is more sensitive to reduction [122]. Tissue factor contains a surface-exposed allosteric disulfide bond in the membrane-proximal fibronectin Type III domain [123]. This allosteric disulfide bond stabilizes the carboxyl-terminal domain involved in interactions with coagulation factors VII and X on a PS-rich surface to trigger coagulation [124,125]. PDI can modify this allosteric disulfide bond by reduction, S-nitrosylation, and glutathionation [124]. Both prothrombin and fibrinogen also undergo carbonylation under oxidative stress [126]. Fibrinogen crosslinks αIIbβ3 integrin to aggregate platelets and, upon cleavage by thrombin, polymerizes into fibrin fibrils. It can be oxidized at multiple sites, reducing the rate of fibrin polymerization [127] and the stability and strength of the fibrin network [128,129]. Oxidized fibrinogen also suppresses platelet aggregation induced by collagen, VWF, and adenosine diphosphate [127]. The prothrombotic genetic variants factor V Leiden and prothrombin G20210A have been associated with PE [130,131] and HELLP syndrome [132,133] but the association has not been consistently detected [134,135,136]. There is no report on oxidative modification of both factors in PE or other settings.

In contrast, the impact of oxidation on fibrinolysis is far less known and remains controversial. On the one hand, oxidized fibrin serves as a better substrate for plasmin to cleave during fibrinolysis and enhances the activation of pro-urokinase [137]. On the other hand, hydroxyl radicals generated from ferric ions convert fibrinogen to aberrant fibrin clots that are resistant to fibrinolytic degradation [138]. Leukocyte-derived oxidant chloramine-T, which preferentially oxidizes methionine residues, reduces the activity of tissue plasminogen activator by 40% [139]. Physiological concentrations of chloramines at 1 mM or higher can oxidize fibrinogen and coagulation factors V, VIII, and X in a dose-dependent manner to reduce their activities and inhibit platelet aggregation [140]. The inhibitory effects of chloramines can be blocked by methionine, cysteine, or ascorbic acid but not by mannitol, superoxide dismutase, or catalase.

Oxidative stress modifies not only proteins but also lipids [141]. Lipid peroxidation could regulate hemostasis during pregnancy through at least two pathways. First, membrane lipids such as PS, which initiates and propagates both intrinsic and extrinsic coagulation, are highly sensitive to peroxidation [142]. Enzymatically oxidized phospholipids restore hemostasis in humans and mice with pathological bleeding, suggesting that lipid peroxidation enhances the procoagulant activity of these anionic phospholipids. Oxidized PS has indeed been shown to enhance the formation of extrinsic tenase, intrinsic tenase, and prothrombinase [143], resulting in a hypercoagulable state. Second, patients with PE often develop dyslipidemia, with increased levels of serum triglyceride, decreased levels of high-density lipoprotein (HDL), reduced sizes of low-density lipoprotein (LDL), and a reduced ratio of LDL cholesterol to apolipoprotein B (LDLc–apo B) in the third trimester [144]. These smaller LDL particles are more susceptible to oxidation [145]. In one report, women with high concentrations of oxidized LDL (≥50 U/L) had a 2.9-fold increased risk of PE over those with less oxidized LDL (95% CI 1.4–5.9) [146].

## 7. Antioxidant Therapies in PE

### 7.1. Endogenous Antioxidants

The primary defense against oxidative stress are endogenous antioxidants such as SOD, catalase, and glutathione peroxidase that neutralize ROS. The level of SOD in the normal placental villous tissues increases from 8 weeks to 20 weeks of gestation [147], presumably to enhance antioxidant defense. The synthesis and enzymatic activity of all three enzymes are lower in placentas from PE patients than those from women with normotensive pregnancies [148,149,150]. Both SOD and catalase are also lower in the blood of PE patients than in that of normotensive pregnant women [151]. The rate-limiting enzyme heme oxygenase-1 (HO-1) catalyzes the potent oxidant heme into carbon monoxide, biliverdin, and free iron, thus serving as an antioxidant against cellular stresses like hypoxia and inflammation. The expression of HO-1 is reduced in placentas from PE patients [152,153]. Nuclear factor erythroid 2-related factor 2 (Nrf2) is a transcription activator that is expressed primarily in cytotrophoblast cells and regulates the expression of antioxidant proteins in these cells. Its expression is upregulated in the placentas of PE patients compared with gestation-matched normotensive subjects [154]. The chemical element selenium is an essential component of the antioxidants glutathione peroxidase and thioredoxin reductase. A meta-analysis by Xu et al. [155] reported that patients with PE have lower levels of plasma selenium than healthy pregnant women.

### 7.2. Therapeutic Antioxidants

Nanoscale selenium (Nano-Se) has been shown to ameliorate hyper-homocysteinemia-induced endothelial injury by inhibiting mitochondrial oxidative damage and apoptosis [156]. Selenium could also protect trophoblast cells from mitochondrial oxidative stress and ROS-mediated apoptosis [157,158], reducing the rate of PE [155]. Women who received 200 mg daily of CoQ10, a proton transfer agent in the mitochondrial electron transport chain, between 20 weeks of gestation and delivery, reduced their risk of PE compared with placebo controls [159]. CoQ10 has also been shown to improve specifically endothelial function [160]. Ergothioneine reduces the risk of PE [161], primarily by mitigating iron-induced oxidative stress. In a rat model of PE, ergothioneine ameliorated hypertension, reduced mitochondrial-specific H_2_O_2_ in the kidneys, and increased the fetal weight [162].

Both antiplatelet aspirin and anticoagulant heparin have been increasingly recommended for women at high risk of preeclampsia [163]. Apart from its conventional antithrombotic activity, aspirin has a modest ability of scavenging superoxide and it also contributes to the release of NO from the endothelium, which might result from the direct acetylation of eNOS [164]. Aspirin-triggered lipoxins inhibit the NADPH oxidase-mediated generation of ROS and nitrotyrosine in the endothelial cells [165,166]. Heparin has also been reported to reduce the plasma level of peroxides and increase the activity of SOD and catalase in red blood cells [167]. Low molecular weight heparin is shown to protect endothelial cells exposed to H_2_O_2_ [168]. However, it is not known whether aspirin and heparin reduce oxidative stress directly or by improving the systemic inflammation and tissue ischemia associated with the prothrombotic state of PE patients. Both the antioxidants pyrrolidine dithiocarbamate and N-acetylcysteine reduce the procoagulant activity of TNF-α-stimulated endothelial cells [72]. Antioxidants and NOX inhibitors significantly reduce platelet activation, aggregation, and thrombus formation [79,169].

## 8. Conclusions

Oxidative stress plays a physiological role in placenta development but it can also cause placental pathologies and systemic conditions that initiate or propagate PE (Figure 2). Oxidative stress modifies the cells and molecules involved in all four components of hemostasis, resulting in hypercoagulable and prothrombotic states. These oxidative modifications are well documented for their biochemistry and cellular impacts but remain poorly understood for their specific roles in the pathogenesis of PE, especially regarding their impact on hemostasis in the condition of PE. Elucidating the role of oxidative stress in the pathogenesis of the PE-induced prothrombotic state could lead to new predictive markers and therapeutic targets for this severe pregnancy complication.

## Figures and Tables

**Figure 1 antioxidants-09-01139-f001:**
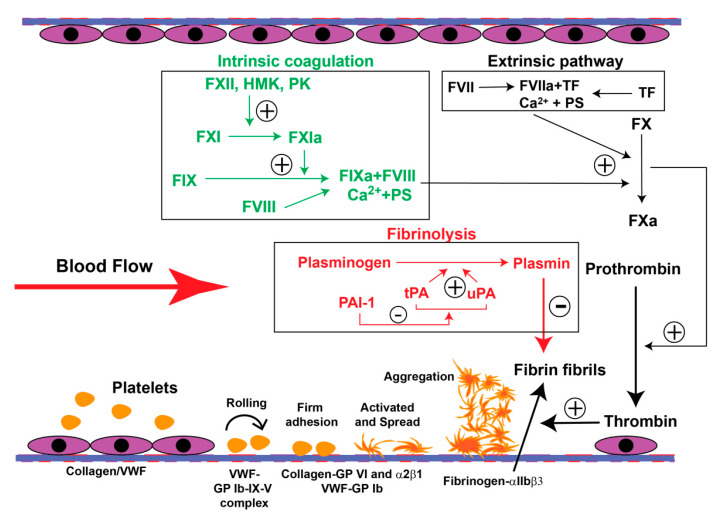
Schematic illustration of human hemostasis. Hemostasis begins at the site of vascular injury. Platelets first adhere to the exposed subendothelium and, through multiple ligand–receptor interactions, are activated and aggregate by fibrinogen crosslinking to form a plug to seal the wound. Anionic phospholipids exposed on the surface of activated platelets also trigger coagulation through a series of enzymatic actions of coagulation factors (e.g., inactive factor X (FX) to activated FX (Fxa)), resulting in the eventual activation of thrombin, which cleaves fibrinogen to generate fibrin fibrils that crosslink and stabilize the platelet plug to arrest bleeding. The fibrin fibrils then activate the fibrinolysis pathway that generates plasmin to cleave these fibrils to re-establish circulation. VWF: von Willebrand factor; TF: tissue factor; PS: phosphatidylserine; HMK: high molecular weight kininogen; PK: prekallibrein; tPA: tissue plasminogen activator; uPA: urokinase.

**Figure 2 antioxidants-09-01139-f002:**
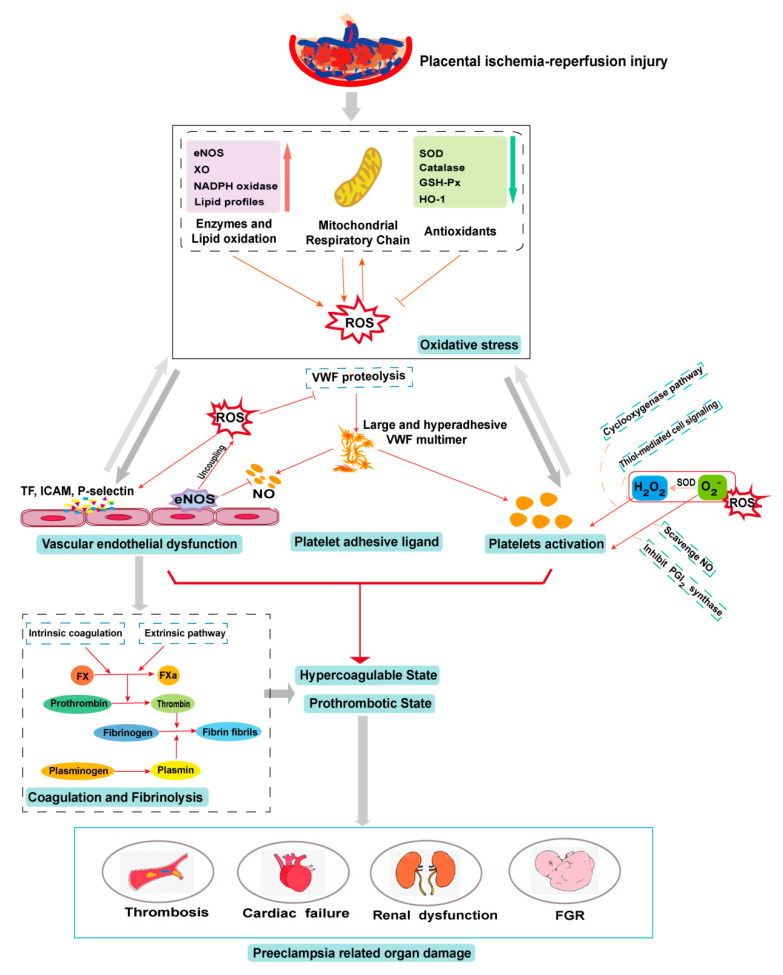
Pathogenesis of the preeclampsia (PE)-associated prothrombotic state. Placental ischemia–reperfusion injury induces mitochondrial dysfunction and oxidative stress and reduces the antioxidative defense, producing excessive reactive oxygen species (ROS). This oxidative stress propagates placental injury and also disseminates to systemic circulation, where it injures endothelial cells and activates platelets to express procoagulant activities and to produce prothrombotic and proinflammatory extracellular vesicles. The procoagulant cells and molecules initiate and propagate the hypercoagulable and prothrombotic states that result in arterial and venous thrombosis, renal dysfunction, and fetal growth restriction (FGR). eNOS: endothelial nitric oxide synthase; XO: xanthine oxidase; SOD: superoxide dismutase; GSH-Px: glutathione peroxidase; HO-1: heme oxygenase-1; TF: tissue factor; ICAM: Intercellular Adhesion Molecule; NO: nitric oxide; PGI_2_: Prostaglandin I_2_.

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
