# Peer review of "Oxidative Stress and Preeclampsia-Associated Prothrombotic State"

_antioxidants, 2020, doi:10.3390/antiox9111139_

Round 1

Reviewer 1 Report

The manuscript entitled: "The oxidative stress and PE-associated prothrombotic state" is well-written review paper. 

In the manucript the Authors describe placental develpoment in pre-eclampsia as well as the pathogenesis of pre-eclampsia. 

Furthermore, the Authors mad e review the literature about pregnancy associated systemic hyper-coagulable state and pre-eclampsia associated prothrombotic state. 

Finally, the write about the role of the oxidative stress and the use of anti-oxidant therapy in pre-eclampsia. 

Author Response

We thank the reviewers for their thoughtful and concise comments and suggestions, and have modified the manuscript accordingly. Individual responses to these comments are discussed in the following sections. Major changes made to address these comments are marked in yellow in the revised manuscript. New references are also included to support the new discussions.

Response to comment by the first reviewer Question:

Comments:

The manuscript entitled: "The oxidative stress and PE-associated prothrombotic state" is well-written review paper. 

In the manucript the Authors describe placental develpoment in pre-eclampsia as well as the pathogenesis of pre-eclampsia. 

Furthermore, the Authors mad e review the literature about pregnancy associated systemic hyper-coagulable state and pre-eclampsia associated prothrombotic state. 

Finally, the write about the role of the oxidative stress and the use of anti-oxidant therapy in pre-eclampsia. 

Response:

We thank the reviewer for the evaluation for this manusript and have also added some information in the revised manuscript markerd in yellow. The revision may has further significantly improved the manuscript.

Reviewer 2 Report

Preeclampsia is a leading cause of fetal and maternal mortality worldwide. The authors submitted their work with the title of Oxidative Stress and Preeclampsia-Associated Prothrombotic State. It is in the focus of interest as there is no accepted treatment for prevention the development of preeclampsia only cesarian section is the final solution.

The manuscript is interesting. Somehow the authors did not give information about the collection of the data. Please insert a section related to the data collection (search keywords, used data base, how many publication did you receive during the search, etc.).

I miss a section related to the genetics of preeclampsia as it is known that Factor V Leiden mutation and prothrombin mutations are more frequent in PE and HELLP syndrome.

It would be interesting to see the differences related to prothrombotic state in PE and HELLP syndrome. Please discuss it.

Author Response

We thank the reviewers for their thoughtful and concise comments and suggestions, and have modified the manuscript accordingly. Individual responses to these comments are discussed in the following sections. Major changes made to address these comments are marked in yellow in the revised manuscript. New references are also included to support the new discussions.

Response to comment by the second reviewer Question:

Comment 1: the authors did not give information about the collection of the data. Please insert a section related to the data collection (search key words, used database, how many publications receive during the search.

Response: The information as requested by the reviewer is now included (page 2, lines 65-70, marked in yellow).

Comment 2: I miss a section related to the genetics of preeclampsia as it is known that Factor V Leiden mutation and prothrombin mutations are more frequent in PE and HELLP syndrome.

Response: The reviewer raised a very important question. Both the Factor V Leiden and prothrombin G20210A variants have been associated with preeclampsia and HELLP syndrome in some studies, but not in others. There is no report on the oxidative modification of both factors in PE or other settings. This information is now provided on pages 10-11 (lines 375-378, marked in yellow).   

Comment 3: It would be interesting to see the differences related to prothrombotic state in PE and HELLP syndrome. Please discuss it.

Response: Patients PE and those with HELLPs are known to develop a hypercoagulable state with similar causal factors. However, it is not known whether the hypercoagulable state differs in level and nature between these two closely related conditions. There is no published report that characterizes the similarity and difference between PE- and HELLPs-induced hypercoagulable states. The oxidative stress is also not specifically compared between the two conditions. We discuss this on page 5 (lines 161-174, marked in yellow)

Comment 4: English language and style are fine/minor check required.

Response: The manuscript has gone through professional editing.

Reviewer 3 Report

Major comments:

The review is well written, in correct and fluent English, and I really don’t have any important suggestion or comments to the authors.

Minor comments:

The only question that bothers me is who is this review for?  Readers of this Journal should be doctors of various specialties and professionals not necessarily of the medical profession. What they have in common is an interest in oxidative and redox processes and antioxidants in biology and medicine. For me personally, reading the text was clear and I followed the story without any problems; probably because I am a gynecologist and subspecialist in maternal-fetal medicine. Therefore, I would suggest authors to think about expanding text with few more sentences just to better present the problem to non-gynecological readers. Maybe the very beginning should be the best part for it. To be specific, I think it should be emphasized that syncytiotrophoblast is essentially apoptotic tissue. This means that the outer membrane side, the one that is in direct contact with the intervillous space and the flow of maternal blood, has the typical membrane structure of each cell in apoptosis. Negatively charged phospholipids on syncytiotrophoblast membrane are presented on the outside, and positively charged phospholipids on the inside - the opposite is the case in cells without induced apoptosis. Such a structural specificity makes the membrane highly thrombogenic even in its physiology.

Literature example:

Pittoni V, Isenberg D. Apoptosis and antiphospholipid antibodies. Semin Arthritis Rheum. 1998;28(3):163-78.

When considering and elaborating about the etiopathogenesis of preeclampsia, early and late form should be distinguished. Their origins are similar, but not completely the same. First one is in direct relation to spiral arteries remodeling process, and the other one has different causes.

Literature example:

Burton GJ, et al. Pre-eclampsia: pathophysiology and clinical implications. BMJ. 2019;366:l2381.

I would suggest authors to point out that preeclampsia, which involves a systemic inflammatory response in the mother, occurs when fetal element, parts of placental villi or trophoblast enter into her systemic circulation provoking systemic inflammatory and immune response. Namely, normal regeneration of trophoblast implies its apoptotic decomposition and flowing “garbage” into mother's circulation in the form of inflammatory inert packages called syncytial knots. Macrophages in the capillary system of mother's lungs normally eliminate them. If placenta releases smaller parts of trophoblast (nano and micro vesicles) into mother's bloodstream, they can pass through the capillary system of the mother's lungs and provoke a systemic inflammatory response by entering the systemic circulation. Preeclampsia as the syndrome is only one of possible clinical sequels.

Literature example:

Chamley LW, et al. Review: where is the maternofetal interface? Placenta. 2014;3:74-80.

It may be a good idea to point out how changes in concentration of some pregnancy related factors can be used in early diagnosis and risk calculation of later preeclampsia onset (Placental growth factor, Soluble Flt-1 and their ratio, than Vascular endothelial growth factor- VGEF, soluble endoglin, PAPP-a, ....

Literature examples:

Zeisler H, et al. Predictive Value of the sFlt-1:PlGF Ratio in Women with Suspected Preeclampsia. N Engl J Med. 2016 ;374(1):13-22.

Serra B, et al. A new model for screening for early-onset preeclampsia. Am J Obstet Gynecol. 2020;222(6):608.e1-608.e18.

Jiang M, Lash GE, Zhao X, Long Y, Guo C, Yang H. CircRNA-0004904, CircRNA-0001855, and PAPP-A: Potential Novel Biomarkers for the Prediction of Preeclampsia. Cell Physiol Biochem. 2018;46(6):2576-2586.

As a potential reader of this paper when published and clinician in the same time, I would like to read at least a few sentences about therapeutic options as part of the discussion in the sense of this paper major theme (Oxidative Stress & Preeclampsia as a Prothrombotic Stat). Especially about heparin as the anticoagulant and immunomodulator in the same time, and eventually use of statins. It is well known that heparin can improve apoptotic syncytiotrophoblast rejuvenation. I don’t find it crucial to this paper, but maybe would add a spice of clinical note and real life applicability.

Literature examples:

- Hills FA, et al. Heparin prevents programmed cell death in human trophoblast. Mol Hum Reprod. 2006 Apr;12(4):237-43.

- Katsi V, et al. The Role of Statins in Prevention of Preeclampsia: A Promise for the Future? Front Pharmacol. 2017 May 5;8:247.

Author Response

We thank the reviewers for their thoughtful and concise comments and suggestions, and have modified the manuscript accordingly. Individual responses to these comments are discussed in the following sections. Major changes made to address these comments are marked in yellow in the revised manuscript. New references are also included to support the new discussions.

Response to comments by the third reviewer

Comment 1: The only question that bothers me is who is this review for?  Readers of this Journal should be doctors of various specialties and professionals not necessarily of the medical profession. What they have in common is an interest in oxidative and redox processes and antioxidants in biology and medicine. For me personally, reading the text was clear and I followed the story without any problems; probably because I am a gynecologist and subspecialist in maternal-fetal medicine. Therefore, I would suggest authors to think about expanding text with few more sentences just to better present the problem to non-gynecological readers. Maybe the very beginning should be the best part for it. 

Response: We thank the reviewer for the suggestion and have briefly discussed our goal in writing this review. We focus this review on understanding the link between prothrombotic state and oxidative stress, which are very common and closely related in the condition of PE (page 2, lines 59-64).

Comment 2: I think it should be emphasized that syncytiotrophoblast is essentially apoptotic tissue. This means that the outer membrane side, the one that is in direct contact with the intervillous space and the flow of maternal blood, has the typical membrane structure of each cell in apoptosis. Negatively charged phospholipids on syncytiotrophoblast membrane are presented on the outside, and positively charged phospholipids on the inside - the opposite is the case in cells without induced apoptosis. Such a structural specificity makes the membrane highly thrombogenic even in its physiology.

Response: We thank the reviewer for the excellent suggestion and have included a brief discussion regarding the apoptotic nature of syncytiotrophoblast and its relation to pregnancy-induced hypercoagulable state (pages 4, lines 130-148, marked in yellow).

Comment 3: When considering and elaborating about the etiopathogenesis of preeclampsia, early and late form should be distinguished. Their origins are similar, but not completely the same. First one is in direct relation to spiral arteries remodeling process, and the other one has different causes.

Response: As suggested by the reviewer, we have now discussed the early-onset and late-onset PF in the Pathogenesis of PE section (page 4, lines 117-124).

Comment 4: I would suggest authors to point out that preeclampsia, which involves a systemic inflammatory response in the mother, occurs when fetal element, parts of placental villi or trophoblast enter into her systemic circulation provoking systemic inflammatory and immune response. Namely, normal regeneration of trophoblast implies its apoptotic decomposition and flowing “garbage” into mother's circulation in the form of inflammatory inert packages called syncytial knots. Macrophages in the capillary system of mother's lungs normally eliminate them. If placenta releases smaller parts of trophoblast (nano and micro vesicles) into mother's bloodstream, they can pass through the capillary system of the mother's lungs and provoke a systemic inflammatory response by entering the systemic circulation. Preeclampsia as the syndrome is only one of possible clinical sequels.

Response: We thank the reviewer for this important suggestion. Our recent studies have indeed shown that placenta-derived extracellular vesicles that express syncytin and the anionic phospholipid phosphatidylserine are released into the maternal circulating, where they cause systemic hypercoagulable and inflammatory states, and vascular permeability (the root cause of proteinuria found in patients with PE), and hypertension [1,2]. This was discussed in the original manuscript (pages 6, now lines 189-200, marked in yellow) and is now expanded to include information suggested by the reviewer (page 4, lines 129-147, marked in yellow). We address comment 2 and 4 from the reviewer in the same paragraph because we believe that the issues are closely related.   

  1. Han, C.; Wang, C.; Chen, Y.; Wang, J.; Xu, X.; Hilton, T.; Cai, W.; Zhao, Z.; Wu, Y.; Li, K., et al. Placenta-derived extracellular vesicles induce preeclampsia in mouse models. Haematologica 2020, 105, 1686-1694, doi:10.3324/haematol.2019.226209.
  2. Chen, Y.; Huang, P.; Han, C.; Li, J.; Liu, L.; Zhao, Z.; Gao, Y.; Qin, Y.; Xu, Q.; Yan, Y., et al. Association of Placenta-Derived Extracellular Vesicles with Preeclampsia and Associated Hypercoagulability: A clinical observational study. BJOG : an international journal of obstetrics and gynaecology 2020, 10.1111/1471-0528.16552, doi:10.1111/1471-0528.16552.

Round 2

Reviewer 2 Report

I accept the answers for my comments.